# Non-Destructive Monitoring via Electrochemical NADH Detection in Murine Cells

**DOI:** 10.3390/bios12020107

**Published:** 2022-02-10

**Authors:** Ju Kyung Lee, Han Na Suh, Sung Hoon Yoon, Kyu Hong Lee, Sae Young Ahn, Hyung Jin Kim, Sang Hee Kim

**Affiliations:** 1Department of Medical IT Convergence, Kumoh National Institute of Technology, Gumi 39177, Korea; chejueyes@kumoh.ac.kr; 2Korea Institute of Toxicology, Jeongeup 56212, Korea; hanna.suh@kitox.re.kr (H.N.S.); seonghoon.yoon@kitox.re.kr (S.H.Y.); khlee@kitox.re.kr (K.H.L.); 3Department of Human and Environmental Toxicology, University of Science & Technology, Daejeon 34113, Korea; 4NDD Inc., Gumi 39253, Korea; sahn@ndd-inc.com; 5Fuzbien Technology Institute, Rockville, MD 20850, USA; 6Digital Health Care Research Center, Gumi Electronics and Information Technology Research Institute (GERI), Gumi 39253, Korea; hjkim745@geri.re.kr

**Keywords:** electrochemical amperometry, surface modification, screen-printed electrode (SPE), polyhexamethylene guanidine-phosphate (PHMG-p), continuous monitoring

## Abstract

Nicotinamide adenine dinucleotide (NADH) is an important cofactor involved in metabolic redox reactions in living cells. The detection of NADH in living animal cells is a challenge. We developed a one-step monitoring method for NADH via an electrocatalytic reaction that uses a surface-modified, screen-printed electrode (SPE) having a redox active monolayer 4′-mercapto-N-phenlyquinone diamine (NPQD) formed by a self-assembled monolayer (SAM) of an aromatic thiol, 4-aminothiophenol (4-ATP). This electrode has a limit of detection (LOD) of 0.49 μM and a sensitivity of 0.0076 ± 0.0006 μM/μA in cell culture media, which indicates that it retains its selectivity. The applicability of this NADH sensor was demonstrated for the first time by cell viability monitoring via NADH-sensing in cell culture supernatants.

## 1. Introduction

Nicotinamide adenine dinucleotide (NADH) is the most well-known biomarker of the redox state of a cell. Several studies have reported that a deficiency of NADH causes a metabolic state problem due to a lack of ATP because ATP production depends on the redox state of coenzymes and NADH. Additionally, mitochondrial dysfunction characterized by inefficient ATP production contributes to neurodegenerative diseases, such as Alzheimer’s, Parkinson’s, and Huntington’s diseases, cardiovascular disease [1,2], diabetes and metabolic syndrome [3,4,5], and lung diseases [6,7]. For these reasons, intracellular NADH has a diagnostic potential as a biomarker for cellular redox reactions, energy production, and mitochondrial functions; therefore, it is important to quantify NADH correctly in in vitro study.

Many studies present various quantification methods for NADH because it exists in cells at a higher concentration than other coenzymes, and its quantification is not affected significantly by blood [8]. Additionally, Adriouch et al. reported that NAD^+^ released during inflammation participates in T-cell homeostasis by inducing the ART2-mediated death of naive T-cells in biogel polyacrylamide bead-induced lung inflammation [9]. The classical method for the determination of NADH is an optical assay using absorbance [10] or fluorescence [11] and a colorimeter [12]. Some classical methods are robust, standardized analytical methods to determine NADH, but they require a colorimeter, which occupies a great deal of space and has a high sample volume (~200 μL) and a high cost per test. An electrochemical (EC) biosensor for NADH has emerged as an alternative to the classical method because it has several advantages, such as convenience and a short analysis time; furthermore, it uses a small sample volume while maintaining high sensitivity and selectivity. These types of sensors use the oxidation/reduction reaction of the NADH/NAD^+^ couplet at a specific potential.

In this research, we developed a screen-printed electrode (SPE) as a substrate, and 4′-mercapto-N-phenylquinone diamine (NPQD) as an electrocatalyst was immobilized by double-step electrochemical functionalization. This NPQD layer can be constructed via the functionalization of 4-aminothiophenol (4-ATP), which was previously immobilized on the Au layer by covalent bonding based on Au-thiol bonding as initially developed by the Takeo group [13]. Although NPQD functionalization is similar, we selected a different electrochemical functionalization method and used a different substrate as the SPE. The limit of detection (LOD) and sensitivity was improved compared with the previous result. Also, we investigated the electrocatalytic reaction in cell culture media to detect NADH in viable cells and observed NADH level by inserting toxicological material to induce cell death. As mentioned above, NADH produced in the mitochondria is an important biomarker for metabolic activity and mitochondrial function, and it is also useful as a biomarker for monitoring living cells because mitochondria lose their functionality after cell death. The reliability of the electrochemical results was compared with that of a conventional WST-1 (water-soluble tetrazolium salt-1) assay, and the two reliabilities were found to be similar. Finally, we challenged the NADH electrochemical sensor with cell culture medium and observed that it performed well in further ex vivo experiments. PHMG-p (Polyhexamethylene guanidine-phosphate) was used as toxic material which induced cell death, and the NADH was quantified and compared via a control model. To the best of our knowledge, this study is the first attempt to quantify NADH electrochemically in vitro, and we believe that this sensor can be used for NADH detection for disease diagnosis and the continuous monitoring of mitochondrial function.

## 2. Materials and Methods

### 2.1. Chemicals

A premixed WST-1 cell proliferation assay kit was purchased from Takara Bio Inc. (Katsu, Japan). RPMI 1640 medium and penicillin-streptomycin were purchased from Life Technologies Co. (New York, NY, USA). Polyhexamethylene guanidine-phosphate (PHMG-p) was provided by SK chemicals (Seongnam, Korea). Potassium ferricyanide [K_3_Fe(CN)_6_], 4-aminothiophenol (4-ATP), 100 mM and 10 mM of Dulbecco’s phosphate buffered saline (DPBS), Tween 20, and absolute ethanol were purchased from Sigma-Aldrich (St. Louis, MO, USA) and used without further purification. All reagents used in the investigation were of analytical grade.

### 2.2. Apparatus and Electrode

Commercial screen-printed electrodes (SPE, Model No: DRP 220 AT, Φ = 4 mm), including a Au working electrode (WE) with a surface area of 12.56 mm^2^, were purchased from the Metrohm DropSens Co. (Oviedo, Spain). This electrode includes a Au counter electrode (CE) and a silver pseudo-reference electrode (RE). Chronoamperometry (CA) and cyclic voltammetry (CV) measurements were conducted with a multi-channel potentiostat obtained from CH instruments (Texas, TX, USA, Model No: CH 1030C) and Wismar Co. Ltd. (Daejeon, Korea, Model No: WIZECM-1200Premium). Electrochemical impedance spectroscopy (EIS) was carried out with a pocketSTAT from Ivium Technologies B.V. (AJ Eindhoven, The Netherlands). All electrochemical (EC) measurements were conducted at room temperature in a Faraday cage.

### 2.3. Surface Modification of the SPE

Prior to assembly, the SPE was pretreated by placing a 50 µL drop of a 10 mM H_2_SO_4_ solution on it, and cyclic voltammetry was performed from 0 V to 1.8 V at a scan rate of 100 mV/s to remove dust. The SPE was then washed with DI water and dried with nitrogen. After drying in a stream of N_2_, the 4-ATP self-assembled monolayer (SAM) was prepared on the SPE by incubating the electrode in 10 mM 4-ATP dissolved in absolute ethanol for 2 h at room temperature as mentioned by the Takeo group [13]. Then, the SPE was washed with absolute ethanol for 1 min and washed again with 0.05% Tween 20 in 10 mM DPBS (pH 7.2) to remove the remaining chemicals. A 4′-mercapto-N-phenylquinone diamine (NPQD) layer was generated on the Au electrode by a two-step electrochemical surface modification. After drying in a stream of N_2_, 50 µL of 100 mM DPBS (pH 7.2) was dropped on the SPE, and CV was performed by applying a potential between 0.8 V and −0.4 V 30 times. After washing the electrode using 10 mM DPBS (pH 7.2), a CV step similar with that conducted in the previous step was performed again by changing the 100 mM DPBS to 10 mM DPBS. After a surface modification, the electrode was washed with 0.05% Tween 20 in 10 mM DPBS and kept in 10 mM DPBS.

### 2.4. In Vitro Studies

#### 2.4.1. Cell Culture and WST-1 Viability Assay

Human epithelial (A549) cells from the American Type Culture Collection (Manassas, VA, USA) were cultured to confluence in culture media (pH 7.2) with 5% FBS and 100 IU/mL penicillin–streptomycin at 37 °C in a humidified atmosphere containing 95% air and 5% CO_2_. Suspensions of A549 Cells (2 × 10^5^ cells/well) were transferred to separate well plates after an overnight incubation at 37 °C in 5% CO_2_. The cytotoxicity of PHMG-p was evaluated versus concentration and time. To investigate the cytotoxicity of the PHMG-p, cells were seeded in a 6-well plate and exposed to increasing concentrations of PHMG-p (i.e., 1.0, 2.0, 3.0, 4.0, and 12.5 µg/mL) for 6, 12, or 24 h. On another plate, the time course effect of PHMG-p was investigated by seeding cells, exposing them to 4 and 8 µg/mL of PHMG-p, and monitoring them for 6, 12, 24, or 36 h. Three replicates were used at each concentration. Cells not treated with PHMG-p were used as the positive control, and media (no cells) was used as the negative control. In both experiments, cell viability was monitored by a WST-1 assay. Prior to adding the WST-1 reagent to the plate, 200 µL of the supernatant was extracted from each plate and placed in a 1.5 mL microtube to quantify the NADH in the cell supernatant by an electrocatalytic reaction. After adding the WST-1 to the plate, the plate was placed in an incubator at 37 °C and 5% CO_2_ for 1 h. The absorbance of each well was measured at 450 nm in a microplate reader. The relative cell viability percentage in each group was calculated by comparing the cell viability in each group to that of the control group.

#### 2.4.2. Quantification of NADH in Cell Supernatants

To quantify the NADH in the cell culture supernatants, 50 µL of the supernatant extracted as described in the previous section was added dropwise to the NPQD-modified electrode. NADH quantification was performed within 10 min of extraction to prevent contamination. Three replicates were used at each concentration, non-treated PHMG-p cells were used as the positive control group, and media (no cells) were used as the negative control. As previously mentioned, we applied a potential of 0.7 V and read the current after 20 s, when it had achieved a stable state.

### 2.5. Statistical Analysis

All values were expressed as the mean ± standard deviation. Statistical analyses were performed using a two-tailed *t*-test. Statistical significance was defined as *p* < 0.05. All assays were run five times and mean and standard deviation were calculated at each concentration to generate the calibration curve. Each replicate was measured with a new screen-printed electrode. The analyte NADH was newly made at each time measurement to maintain fresh conditions. Linear curve fitting was performed with the Origin 8.0 program.

## 3. Results and Discussion

Many important aspects of the metabolic state of the mitochondria can be evaluated by monitoring the NADH in the cell in terms of energy production and intracellular oxygen levels [14]. MTT (3-(4,5-dimethylthiazol-2-yl)-2,5-diphenyltetrazolium bromide) and WST-1 (water soluble tetrazolium salt-1) analysis is widely used for cell viability via NADH detection (Figure 1a). A549 cells were seeded in 6-well or 12-well plates and incubated for 24 h. Additionally, the cells were treated with PHMG-p. After the cells were incubated for 6, 12, and 24 h to observe the time effect induced by PHMG-p, the supernatant was collected. A collected sample was used for WST-1, MTT, and electrocatalytic analysis. We further investigated whether our electrocatalytic sensor could detect NADH to monitor mitochondrial dysfunction in an in vitro cell culture system. To achieve this, the NADH electrocatalytic sensor was used in a cell viability assay to determine the number of viable cells after a defined incubation period [15]. Figure 1b shows a schematic illustration of the formation of the NPQD layer for the electrocatalytic reaction of NADH. The substance 4′-mercapto-N-phenylquinone diamine (NPQD) was chosen as the electrocatalyst and immobilized by electrochemical functionalization due to its good redox behavior and rigid structure [13]. In addition, it lowers the oxidation potential and enhances the current because diamines are electroactive, easily oxidized, and can transfer two electrons via NADH oxidation to NAD^+^ (one electron transfer).

The NPQD layer can be constructed via the electrochemical functionalization of 4-ATP (4-aminothiophenol) which was previously immobilized by covalent bonding as first developed by the Takeo group. In this paper, NPQD was generated with a similar process but a different electrochemical functionalization method to improve the performance. Figure 1a shows the CV (cyclic voltammogram) data as obtained during the functionalization of 4-ATP in 100 mM PBS buffer (pH 7.2). This result is similar but different from that in previous reports because the cathodic current at 0.55 V and the anodic current at −0.2 V decreased and the reversible redox peak at 0.23 V newly emerged via a repeating CV cycle. However, the redox peak at 0.23 V was not observed, and we speculated that this was due to differences in our proposed system. The SPE consists of Au working, Au counter, and Ag reference electrodes, and 4-ATP was immobilized on the working electrode and on the counter electrode. This caused electrochemical motion in a manner different from that in previous reports because only the Au working electrode, and not the counter electrode, was functionalized. Given that the repeated CV signal in 100 mM PBS buffer is unstable, we performed a “double-step” electrochemical functionalization using a low concentration of PBS buffer (10 mM con, pH 7.4) after a high concentration PBS buffer for stabilization while maintaining other CV parameters. Although the current, approximately corresponding to −0.4 V, did not exhibit significant fluctuation, changes observed in the high concentration were not observed in the 10 mM PBS buffer as shown in Figure 1b.

The NADH-sensing performance of the single-step electrochemical functionalization using 100 mM PBS buffer and double-step electrochemical functionalization sequentially using 100 mM and 10 mM PBS buffer are shown in Figure 1c. The red line is the double-step electrode’s performance, and the slope is 0.41, and the black line is the single-step electrode’s performance, and the slope is 0.22, respectively. We found the slope (sensitivity) was improved two times, compared with the previous method. Also, the LOD value of the double-step polymerized reaction had 494 nM, and a single-step polymerized reaction is 1.4 μM. We found that the LOD and sensitivity were improved compared with previous method. As a result, correct quantitative measurement is possible.

As shown in the Figure 1d, EIS (electrochemical impedance spectroscopy) was used to investigate unmodified 4-ATP- and NPQD-modified Au surfaces at a constant concentration of redox species of Fe(CN)_6_^3−/4−^, and we found a charge-transfer resistance (R_ct_) that was expressed by the diameter of a semicircle in a Nyquist plot of the NPQD-modified surface, which had a minimum value (63.79 kΩ) compared with the unmodified and ATP-modified electrodes (1041 kΩ and 583.4 kΩ, respectively). Additionally, a 45° line in the Nyquist plot indicates a Warburg region of semi-infinite diffusion of a species in the modified electrode, and the NPQD-modified electrode clearly shows a diffusion process governed by the mass transport of the redox molecules from the solution to the electrode. NPQD functionalization was also demonstrated by the contact angle measurement of a water droplet and EDAX analysis because the -NH functional group in NPQD is polarized and hydrophilic. As shown in Figure 1e, the contact angle of the bare electrode was 63.4°, and for the 4-ATP modified electrode, it was 56.9°, a value not much different from that of the bare electrode. However, the NPQD-modified electrode showed a much lower value of 34.9° due to the amine group of NPQD. EDAX images also confirm the stepwise modification [16] (see Appendix A).

Figure 2a–c depicts the cyclic voltammetry and chronoamperometry demonstrating the electrocatalytic activity of the NPQD-Au electrode for NADH oxidation in medium buffer. A cyclic voltammogram from −100 mV to 700 mV shows that the current did not change until 400 mV when NADH was added. However, a dramatic enhancement of anodic current began from the 400 mV higher potential range when NADH was added. This current was caused by NADH oxidation to NAD^+^, which regenerates the diamine, as shown in the schematic in Figure 1. To oxidize NADH to NAD^+^, a potential is required as the driving force, and we found that the minimum potential for oxidation is 400 mV. Next, we constructed a calibration plot of NADH at various potentials found from cyclic voltammetry (400, 500, 600, and 700 mV), and the 600 mV results showed the widest sensitivity range and linear range. Chronoamperometry was used to make a calibration plot of NADH, and a short, fixed potential analysis time (10 s) per sample was required, compared with cyclic voltammetry [17,18]. A plot of the steady current at 10 s vs. the NADH concentration followed the adjusted equation I (nA) = {(−5.67)·[NADH], µM − 13.64} (R^2^ = 0.999), with a range of linearity between 1.8~1000 µM and had a 250 nM of LOD (limit of detection). The LOD value was calculated from the measurements carried out with three different NADH sensors for each concentration. The achieved sensitivity and sensing range was applicable to cell studies and non-clinical animal experiments, such as those conducted on mice, because the known concentrations in animal cells are approximately 0.3 mM [8,19].

The viability obtained by the WST-1 assay depends on the number of live cells, as they use the product formazan, which is generated by mitochondrial dehydrogenases in live cells. However, viability is determined differently with the EC (electrocatalytic) and WST-1 assays. We hypothesized that measured NADH would greatly increase directly after cell death. Cell viability was calculated by the following Equations (1) and (2).
(1)$${\mathrm{Cell}\;\mathrm{viability}}_{\mathrm{EC}}\;\left(\%\right)=100-\left({\mathrm{VF}}_{\mathrm{EC}}\right)=100-\left(\;\frac{I_{s}-I_{c}-I_{0}}{I_{c}-I_{0}}\times 100\right)$$

Equation (1): Cell viability by the electrocatalytic assay: I_s_ is the current value of a sample treated with a specific concentration of PHMG-p, I_c_ is the current value of the control sample that was not treated with PHMG-p, and I_0_ is the current value of the blank. VF_EC_ is the viability factor by electrocatalytic reaction. All current was measured at a 600 mV potential.
(2)$${\mathrm{Cell}\;\mathrm{viability}}_{\mathrm{WST}-1}\;\left(\%\right)=\left(\frac{A_{S}-A_{b}}{A_{c}-{\;A}_{b}}\right)\times 100$$

Equation (2): Cell viability by the WST-1 assay. A_s_ is the optical density at 450 nm of a sample treated with a specific concentration of PHMG-p (cells treated with PHMG-p+WST-1), A_c_ is the optical density of the control sample that was not treated with PHMG-p (the cells were not treated with PHMG-p+WST-1), and A_b_ is the optical density of the blank (WST-1). All measurements were performed in the cell culture medium.

To test this hypothesis, the cytotoxic effect of PHMG on A549 cells was evaluated by comparing the electrocatalytic measurements to the results of the WST-1 conventional cell viability assay (Figure 3). Raw data of electrocatalytic reaction and conventional sensing data is shown at Appendix A. Previously, we already established an animal model of PHMG-p-induced lung inflammatory and fibrotic responses and reported that PHMG-p induces polymorphonuclear cells (PMN) and macrophage-dominant lung inflammation during week 1 and marked lung fibrosis from weeks 2 to 10 via histologic analyses of H&E and Masson’s trichrome stained preparations [20].

First, to understand the effect of PHMG-p concentration on the cells, we performed a cell viability assay with exposure to 1.0, 2.0, 3.0, 4.0, and 12.5 μg/mL PHMG-p. As expected, a decrease in cell viability was observed with an increasing concentration of PHMG-p. It is worth noting that the viability value obtained by the EC assay was similar to that obtained by the WST-1 assay, even though the NADH-sensing approaches in the two assays are different. Another aspect of the EC viability assay is the time–dose effect of PHMG-p. To understand this, the current was measured at 700 mV by using the cell culture media after exposure to 4 μg/mL of PHMG-p. The PHMG-exposure concentration of 4 μg/mL was used because it exhibits the cytotoxic effect more clearly than other concentrations. A comparison with the control revealed that the current value was most significantly increased after PHMG-exposure time (Figure 4) with fixing PHMG-concentration (4 μg/mL). After 6, 12, and 24 h, the current increased by 21.33 (Figure 4a), 26.03 (Figure 4c), and 46.35% (Figure 4e), respectively, compared with the control. This means that increasing exposure time leads to increased cell death. As shown in Figure 3b,d,f, negative current value was increased with increasing exposure time because NADH was generated when cells die. To fully understand the PHMG-p exposure-time effect, cells were exposed to various concentrations of PHMG-p, and the viability was observed after 6, 12, and 24 h. As shown in Figure 3a–d, a decrease in cell viability with an increasing PHMG-p exposure time was observed in both the WST-1 and EC cell viability assays. The 50% lethal concentration (LC 50) of PHMG-p in the WST-1 and the EC assays was 7.76 μg/mL at the 12-h exposure time and 3.38 μg/mL at the 24-h exposure time, respectively (Equations (1) and (2)).

These data suggest that increased concentrations and exposure times of PHMG-p result in reduced cell viability. This viability was measured by the EC assay, and the result was similar to that obtained with the WST-1 conventional assay. The greatest advantage of the EC assay is that cell viability can be measured continuously, as the EC assay does not require chemicals, such as tetrazolium salts; therefore, the samples are not damaged as in the WST-1 conventional assay. To verify these results, continuous monitoring of cell viability was performed by the EC assay after exposure to 4 and 8 μg/mL of PHMG-p (Figure 3d). The cell viability after exposure to both concentrations decreased, but the results for 8 μg/mL showed a lower cell viability than those for 4 μg/mL. The lethal time (LT 50) for 4 μg/mL was 23.24 h, and the LT 50 for 8 μg/mL was 12.41 h. These results are similar with the results shown in Figure 3a–d. Taken together, these data strongly suggest that cell viability decreased with an increasing PHMG-p concentration and exposure time, and that our EC assay has the potential to not only replace the conventional cell viability assay, but also to aid in cytotoxicity studies of a variety of toxic materials as continuous cell viability monitoring is also possible. Table 1 presents the many types of sensors to measure NADH used in the past 5 years [21,22,23,24,25,26,27,28,29,30,31,32,33,34,35,36,37,38]. Our electrocatalytic sensor has the advantages of comprising one simple step, consuming a low volume of sample (50 μL), and having a low LOD. Also, this is the first attempt to use it as a tool to observe the relationship with NADH and cell viability and apply it to toxicological study.

We have demonstrated that changes in the NADH concentration, as determined by our electrocatalytic sensor, in live cells can be used to monitor cytotoxic events caused by the addition of toxic compounds such as PHMG-p, which provides a framework for toxicity studies for the first time. Such a technique permits the sensing of NADH redox signaling without disrupting or destroying the cells to facilitate the investigation of a complex biological fluid. Individual monitoring via group, continuous, or end-point monitoring are some of the advantages of our electrocatalytic sensor which can improve the understanding of the relationship between NADH and cytotoxic effects. Although further studies are required for an accurate understanding of the mechanism of NADH in this study, this approach could establish NADH as a potential and important biomarker of cytotoxicity and the electrocatalytic sensor as an efficient assay for NADH concentration.

## 4. Conclusions

We demonstrated that the developed electrocatalytic sensor enables detection of NADH and can be applied to cell viability with simple devices. To increase the LOD and sensitivity, a double-step polymerized method was used. As a result, the LOD was 0.45 μM and analysis time was 10 s per sample. The sensor was applied to detect cell viability by inserting PHMG-p as toxic matter, and we found that NADH level increased when we inserted a high level of PHMG-p because it caused cell death. This demonstrates that the NADH electrocatalytic sensor will become a powerful tool for cell viability assays and pave the way for toxicological study.

## Data Availability

Not applicable.

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
