# Peer review of "Non-Destructive Monitoring via Electrochemical NADH Detection in Murine Cells"

_biosensors, 2022, doi:10.3390/bios12020107_

Round 1
Reviewer 1 Report
This paper investigated the electrochemical NADH biosensor to detect NADH from the in vitro cultured samples for continuous and simultaneous monitoring of cell viability in vitro. NADH produced in the mitochondria is an essential biomarker for metabolic activity and mitochondrial function, and it is also helpful as a biomarker for monitoring cells’ viability. Therefore, this sensor might benefit NADH detection for disease diagnosis and the continuous monitoring of mitochondrial function, etc. This paper is of interest. However, it requires addressing major and minor concerns to be published in this journal.
- A major concern is how to interpret the cell viability from obtained current values at 600 mV using EC (electrocatalytic) assay, which directly detects NADH using the biosensor? This paper has no direct evidence or data to define a relationship between NADH release by adding various concentrations of PHMG and cell viability measured by WST-1 assay. At least two studies or data are required. The first data will be a comparison between the measured concentration of NADH in the collected samples using both well-defined conventional methods and EC assay (NADH biosensor) to evaluate the performance of the NADH biosensor. Another data should define a relationship between NADH concentration in the collected samples analyzed by the conventional methods and cell viability measured by WST-1 assay. The WST-1 assay does not directly detect the NADH, although the NADH is produced in the mitochondria. Therefore, with current data, we cannot directly convert the NADH concentration to cell viability.
- In line 235, the author said that “cell viability was calculated by the following equation.” But there is no equation information following the sentence.
- In figure 1c, sample legends as 10x 1x sequential and 10x only are not clear. Please change the sample legends in figure 1c. Also, what is the meaning of the result, which was increased slope (sensitivity) of the double-step functionalization two times compared with the singe step functionalization? There is an explanation about the figure 1c result. But there is no discussion about the results.
- There is no reason why the contact angle measurement is needed. Is this measurement used to evaluate NPQD functionalization? If yes, another method such as AFM or Raman or other ways will be better than the contact angle measurement. Also, whether hydrophilicity and polarization of NPQD are affected the sensing performance?
- Is there any reason to select the PHMG to evaluate the cytotoxic effect? Please add a reason why PHMG was used.
- Also, the author said that “As expected, a decrease in cell viability was observed with an increasing concentration of PHMG-p. It is worth noting that the viability value obtained by the EC assay was similar to that obtained by the WST-1 assay,….” on lines 244-247. However, when I see figure 3, the cell viability characterized by EC viability assay was increased in figure 3 a-c. However, in figure 3d, the cell viability was decreased when the concentration of PHMG increased over time. The results contradict each other. It is necessary to check raw data again.
- Statistical analysis method should be provided on figure 4.
- Also, this experiment was performed in the static culture condition and collected samples from each time point. So, I’m not sure whether we can say "continuous and simultaneous cell viability monitoring" or not in the title. Maybe “non-invasive monitoring” will be a better term for the advantages of the biosensor.
- Overall in this manuscript, the discussion is poorly written.
Author Response
from the in vitro cultured samples for continuous and simultaneous monitoring of cell viability in vitro. NADH produced in the mitochondria is an essential biomarker for metabolic activity and mitochondrial function, and it is also helpful as a biomarker for monitoring cells’ viability. Therefore, this sensor might benefit NADH detection for disease diagnosis and the continuous monitoring of mitochondrial function, etc. This paper is of interest. However, it requires addressing major and minor concerns to be published in this journal.
Reviewer 1 Comment 1: A major concern is how to interpret the cell viability from obtained current values at 600 mV using EC (electrocatalytic) assay, which directly detects NADH using the biosensor? This paper has no direct evidence or data to define a relationship between NADH release by adding various concentrations of PHMG and cell viability measured by WST-1 assay. At least two studies or data are required. The first data will be a comparison between the measured concentration of NADH in the collected samples using both well-defined conventional methods and EC assay (NADH biosensor) to evaluate the performance of the NADH biosensor. Another data should define a relationship between NADH concentration in the collected samples analyzed by the conventional methods and cell viability measured by WST-1 assay. The WST-1 assay does not directly detect the NADH, although the NADH is produced in the mitochondria. Therefore, with current data, we cannot directly convert the NADH concentration to cell viability.
Response and Action: We thank the reviewer for his/her criticism. First, we compared the our NADH quantitative assay (electrochemical) with the conventional assay (optical assay). Conventional assay, we used the abcam NAD/ NADH Assay Kit Ⅱ (colorimetric) (ab221821). As shown the figure R1, the range from 1~250 μM, our electrochemical assay results similar with the optical assay result. The linear fitting equation are y=-0.0052x -0.4376 (electrochemical) and y=0.0059 x + 1.17 (conventional), respectively. Although, the direction is opposite, they have the similar slope value.
After observing the conventional and electrochemical assay, we quantified the NADH in standard sample. The standard sample were made by specific concentration of NADH dissolved in the medium. The concentration are 10, 100, 200 μM, respectively. The table R1 is the result of NADH quantification. We find the measured data (electrochemical and conventional assay, both of them) is almost matched with the standard sample’s concentrations. In these results, we prove that our electrochemical sensor could be measured the NADH concentration correctively.
Figure R1. NADH measurement of electrochemical and conventional assay.
|
Electrochemical |
Conventional |
Con (standard) μM |
Sample 1 |
11±0.45 |
9.71±0.45 |
10 |
Sample 2 |
9.43±0.51 |
9.91±0.14 |
100 |
Sample 3 |
198.01±034 |
199.97±0.65 |
200 |
Table 1. NADH quantitative performance of electrochemical and conventional assay using standard sample.
Reviewer 1 Comment 2: In line 235, the author said that “cell viability was calculated by the following equation.” But there is no equation information following the sentence.
Response and Action: We thank the reviewer for his/her criticism. Cell viability equation is mentioned at equation 1. So, we revised the sentence.
Revised to
Cell viability was calculated by the following equation 1-2.
Reviewer 1 Comment 3: In figure 1c, sample legends as 10x 1x sequential and 10x only are not clear. Please change the sample legends in figure 1c. Also, what is the meaning of the result, which was increased slope (sensitivity) of the double-step functionalization two times compared with the singe step functionalization? There is an explanation about the figure 1c result. But there is no discussion about the results.
Response and Action: We thank the reviewer for his/her criticism. Yes. double step polymerized electrode’s sensitivity is improved compared with single step sensitivity. Most of the cell have the μM range of NADH. For example, the averaged NADH concentration in breast cancer cells is 168± 49 μM by measuring the 2 photon autofluorescence dynamics assay [1]. That is why the development of a rapid, cheap, and high accuracy method that maintains sensitivity is still important in the NADH sensor field. In this experiment, the LOD value of double step polymerized reaction has 494 nM and single polymerized reaction is 1.4 μM. The LOD and sensitivity are improved compared with previous method. As a result, correct quantitative measurement is possible. We insert the explanation in the main manuscript.
Insert
Also, the LOD (limit of detection) value of double step polymerized reaction has 494 nM and single polymerized reaction is 1.4 μM. We found that the LOD and sensitivity are improved compared with previous method. As a result, correct quantitative measurement is possible.
Reviewer 1 Comment 4: There is no reason why the contact angle measurement is needed. Is this measurement used to evaluate NPQD functionalization? If yes, another method such as AFM or Raman or other ways will be better than the contact angle measurement. Also, whether hydrophilicity and polarization of NPQD are affected the sensing performance?
Response and Action: We thank the reviewer for his/her criticism. First, contact angle was measured to observe amine functional group. Figure R2 is the 4-ATP (4-aminothiophenol) chemical. They have thiol group and amine group. Gold and thiol group is connected by covalent bonding, so amine group is exposed the electrode’s surface. After, electropolymerization is performed by cyclic voltammetry and NPQD is generated on the electrode’s surface. Both of them, are the amine group and amine is hydrophilic group. That is why we measured the contact angle. We also insert the EDAX and SEM image to support the NPQD functionalization was well defined on the electrode.
Figure R2. 4-aminothiophenol
To support the electrode modification, we performed the SAM and EDAX analysis.
Figure R2. EDAX spectrum of (a) bare, (b) 4-ATP modified, (c) NPQD modified electrode.
Compound %
|
Au |
Bare electrode |
100.00 |
Compound %
|
N |
Au |
4-ATP |
1.72 |
98.28 |
Compound %
|
N |
Au |
NPQD modified |
2.00 |
98.00 |
As shown the figure R2, bare electrode, they are composed of Au, and 4-ATP (they have amine group), so we can observe N compound. Also, NPQD they also have N induced by 4-ATP. Therefore, N element was found. EDAX image support the step wise modification.
Reviewer 1 Comment 5: Is there any reason to select the PHMG to evaluate the cytotoxic effect? Please add a reason why PHMG was used.
Response and Action: We thank the reviewer for his/her criticism.
Polyhexamethylene guanidine (PHMG), a guanidine-based biocide, was used a major ingredients of household products such as shampoos, wet wipes and humidifier disinfectant. PHMG-containing various types of humidifier disinfectant also have been used for several years and resulted in wide exposure in the korean populations. But, according to the Korea Centers for Disease Control and Prevention (KCDC), the fatal pulmonary diseases was first reported and confirmed that humidifer disinfectants had a significant relevance with fatal pulmonary diseases. Since this reports, researchers began investigating the humidifier disinfectant toxicity to assess relations with lung diseases. Then, unintentiaonal inhalation exposure by humidifer disinfectatn to humans caused an unexpected massive fatal pulmonary diseases. But still, studies and risk assessment were needed to clarify causal relationships.
PHMG was a major ingredients of humidifer disinfactant for anti-microbial growth properties and were still considered for applications in industrial areas. previous research demonstrates the occurrence of PHMG-induced fibrotic responses in the lungs. And, various types of respiratory diseases, including asthma, pneumonia and pulmonary fibrosis were related to pulmonary toxicity-induced by PHMG. Therefore, it is necessary to investigate the potential effects of PHMG on lung injury. In this research, Korea institute Toxicology (KIT) performed the toxicity effect induced by PHMG, and observe their effect by single cell model, that is why we used the PHMG as a toxicology material.
We insert the sentences about PHMG effect and their importance in introduction part.
Insert intro
Finally, we challenged the NADH electrochemical sensor with cell culture medium and observed that it performed well in further ex vivo experiments. PHMG-p (Polyhexamethyleneguanidine phosphate) was used as toxic material which induced cell death, and the NADH was quantified and compared via a control model.
Insert in result and discussion part
Previously, we already established an animal model of PHMG-p-induced lung inflammatory and fibrotic responses and reported that PHMG-p induces polymorphonuclear cells (PMN) and macrophage-dominant lung inflammation during week 1 and marked lung fibrosis from weeks 2 to 10 via histologic analyses of H&E and Masson's trichrome stained preparations [23].
Reviewer 1 Comment 6: Also, the author said that “As expected, a decrease in cell viability was observed with an increasing concentration of PHMG-p. It is worth noting that the viability value obtained by the EC assay was similar to that obtained by the WST-1 assay,….” on lines 244-247. However, when I see figure 3, the cell viability characterized by EC viability assay was increased in figure 3 a-c. However, in figure 3d, the cell viability was decreased when the concentration of PHMG increased over time. The results contradict each other. It is necessary to check raw data again.
Response and Action: We thank the reviewer for his/her criticism. We mistake for calculating ell viability. We checked again the raw data, and cell viability by measuring WST-1 and electrocatalytic sensor was shown at Figure 3. Also, raw data was shown at figure S2.
Reviewer 1 Comment 7: Statistical analysis method should be provided on figure 4.
Response and Action: We thank the reviewer for his/her criticism. We insert the statistical analysis in experimental method part.
Insert
Statistical analysis
All values were expressed as the mean±standard deviation. Statistical analyses were performed using a two-tailed t-test. Statistical significance was defined as p < 0.05. All assays were run five times and mean and standard deviation were calculated at each concentration to generate the calibration curve. Each replicate was measured with a new screen-printed electrode. The analyte NADH was newly made at each time measurement to maintain fresh conditions. Linear curve fitting was performed with Origin 8.0 program.
Reviewer 1 Comment 8: Also, this experiment was performed in the static culture condition and collected samples from each time point. So, I’m not sure whether we can say "continuous and simultaneous cell viability monitoring" or not in the title. Maybe “non-invasive monitoring” will be a better term for the advantages of the biosensor.
Response and Action: We thank the reviewer for his/her criticism. Yes. that is right. We want to present that all measurement was performed the daily. That is why we used the ‘continuous monitoring”. However, continuous monitoring means the real time monitoring. In fact, results in this experiment is closed to static assay than dynamic assay same as “real time monitoring”. So, we change the title and remove the “continuous” word. “non-destructive monitoring” is more adequate to support the result in this experiment.
Title, Revised to
“Non-destructive cell viability monitoring via electrochemical NADH detection in murine cells”
Reviewer 1 Comment 9: Overall in this manuscript, the discussion is poorly written.
Response and Action: We thank the reviewer for his/her criticism. We revised whole part of discussion part.
Revised to
These data suggest that increased concentrations and exposure times of PHMG-p result in reduced cell viability. This viability was measured by the EC assay, and the result was similar with that obtained with the WST-1 conventional assay. The greatest advantage of the EC assay is that cell viability can be measured continuously, as the EC assay does not require chemicals such as tetrazolium salts; therefore, the samples are not damaged as in the WST-1 conventional assay. To verify these results, continuous monitoring of cell viability was performed by the EC assay after exposure to 4 and 8 mg/mL of PHMG-p (Figure 3d). The cell viability after exposure to both concentrations decreased, but the results for 8 mg/mL showed a lower cell viability than those for 4 mg/mL. The lethal time (LT 50) for 4 mg/mL was 23.24 h and the LT 50 for 8 mg/mL was 12.41 h. These results are similar with the results shown in Figures 3a–d. Taken together, these data strongly suggest that cell viability decreased with an increasing PHMG-p concentration and exposure time and that our EC assay has the potential to not only replace the conventional cell viability assay but also aid in cytotoxicity studies of a variety of toxic materials as continuous cell viability monitoring is also possible. Table 1 is the many types of sensors to measure NADH from recent 5 years [21-38]. Our electrocatalytic sensor has the advantages of being a simple step, consuming a low volume of sample (50 μL), having a low LOD. Also, it is first attempt used as a tool to observe the relationship with NADH and cell viability. and applicability to toxicological study.
We have demonstrated that changes in the NADH concentration as determined by our electrocatalytic sensor in live cells can be used to monitor cytotoxic events caused by the addition of toxic compounds such as PHMG-p, which provides a framework for toxicity studies for the first time. Such a technique permits the sensing of NADH redox signaling without disrupting or destroying the cells to facilitate the investigation of a complex biological fluid. Individual monitoring via group, continuous, or end-point monitoring are some of the advantages of our electrocatalytic sensor that can improve the understanding of the relationship between NADH and cytotoxic effect. Although further studies are required for an accurate understanding of the mechanism of NADH in this study, this approach could establish NADH as a potential and important biomarker of cytotoxicity and the electrocatalytic sensor as an efficient assay for the NADH concentration.
Reviewer 1 Comment 10: In line 235, the author said that “cell viability was calculated by the following equation.” But there is no equation information following the sentence.
Response and Action: We thank the reviewer for his/her criticism. This comment is same with reviewer 1 comment 2.

Reviewer 2 Report
In this study, a one-step monitoring method for NADH via an electrocatalytic reaction was achieved. This manuscript should be improved.
Specific comments are as follows:
- L55-58. The limitation of previous studies for NADH monitoring is not clear. The research gap and the contribution of this study is not clear.
- L55. A paper should be cited here that “Recently, we developed a screen-printed electrode (SPE)”.
- Scheme 1. The electrocatalytic assay is not clear. The extraction process and the cell were not plotted.
- More physicochemical evidence are needed here that NPQD was successfully modified on the SPE electrode.
- Results from the study should be listed in the table.
- Conclusions. This is a "conclusion" section, not for the list of results. Too long. Kindly reorganize and summarize your content in inductive language. It cannot be extensively discussed here with the citing of a table.
- Graphical abstract. It can be provided.
- Papers cited here should be updated. Only 2 papers published in recent 3 years were cited.
Author Response
reaction was achieved. This manuscript should be improved.
Reviewer 2 Comment 1: L55-58. The limitation of previous studies for NADH monitoring is not clear. The research gap and the contribution of this study is not clear
Response and Action: We thank the reviewer for his/her criticism. We revise the intro part as requested.
Revise to
In this research, we developed a screen-printed electrode (SPE) as a substrate, and 4’-mercapto-N-phenylquinone diamines (NPQD) as an electrocatalyst was immobilized by double step electrochemical functionalization. This NPQD layer can be constructed via the functionalization of 4-aminothiophenol (4-ATP), which was previously immobilized on the Au layer by covalent bonding based on Au-thiol bonding as initially developed by the Takeo group [13]. Although NPQD functionalization is similar, we selected a different electrochemical functionalization method and used a different substrate as the SPE. The limit of detection (LOD) and sensitivity was improved compared with previous result. Also, we investigated the electrocatalytic reaction in cell culture media to detect NADH in viable cells and observed NADH level by inserting toxicological material to induce cell death. As mentioned above, NADH produced in the mitochondria is an important biomarker for metabolic activity and mitochondrial function, and it is also useful as a biomarker for monitoring living cells because mitochondria lose their functionality after cell death. The reliability of the electrochemical results was compared with that of a conventional WST-1 (water soluble tetrazolium salt-1) assay, and the two reliabilities were found to be similar. Finally, we challenged the NADH electrochemical sensor with cell culture medium and observed that it performed well in further ex vivo experiments. PHMG-p (Polyhexamethyleneguanidine phosphate) was used as toxic material which induced cell death, and the NADH was quantified and compared via a control model. To the best of our knowledge, this study is the first attempt to quantify NADH electrochemically in vitro, and we believe that this sensor can be used for NADH detection for disease diagnosis and the continuous monitoring of mitochondrial function.
Reviewer 2 Comment 2: L55. A paper should be cited here that “Recently, we developed a screen-printed electrode (SPE)”.
Response and Action: We thank the reviewer for his/her criticism. We revise the sentence.
Revise to
In this research, we developed a screen-printed electrode (SPE) as a substrate, and 4’-mercapto-N-phenylquinone diamines (NPQD) as an electrocatalyst was immobilized by double step electrochemical functionalization.
Reviewer 2 Comment 3: Scheme 1. The electrocatalytic assay is not clear. The extraction process and the cell were not plotted.
Response and Action: We thank the reviewer for his/her criticism. We insert the sentences about extraction process and the cell.
Insert the sentences
A549 cells were seeded in 6-well or 12 well plates and incubated for 24 hr. And the cells were treated with PHMG-p. After the cells were incubated for 6, 12, 24hour to see the time effect induced by PHMG-p, the supernatant was collected. Collected sample was used for WST-1, MTT and electrocatalytic analysis.
Reviewer 2 Comment 4: More physicochemical evidence are needed here that NPQD was successfully modified on the SPE electrode.
Response and Action: We thank the reviewer for his/her criticism. To support the electrode modification, we performed the SAM and EDAX analysis.
Figure R2. EDAX spectrum of (a) bare, (b) 4-ATP modified, (c) NPQD modified electrode.
Compound %
|
Au |
Bare electrode |
100.00 |
Compound %
|
N |
Au |
4-ATP |
1.72 |
98.28 |
Compound %
|
N |
Au |
NPQD modified |
2.00 |
98.00 |
As shown the figure R2, bare electrode, they are composed of Au, and 4-ATP (they have amine group), so we can observe N compound. Also, NPQD they also have N induced by 4-ATP. Therefore, N element was found. EDAX image support the step wise modification. It was matched with previous result [2].
Reviewer 2 Comment 5: Results from the study should be listed in the table.
Response and Action: We thank the reviewer for his/her criticism. we insert our results in the table 1 which is summarized the experiment.
Insert our result in the table 1.
Reviewer 2 Comment 6: Conclusions. This is a "conclusion" section, not for the list of results. Too long. Kindly reorganize and summarize your content in inductive language. It cannot be extensively discussed here with the citing of a table.
Response and Action: We thank the reviewer for his/her criticism. we modified the “conclusion” part as reviewers requested. Also, table 1 was mentioned at result and discussion part.
Conclusion (modified)
We demonstrated that the developed electrocatalytic sensor enables detection of NADH and applied to cell viability with simple devices. To increase the LOD and sensitivity, double step polymerized method was used. As a result, the LOD was 0.45 μM and analysis time is 10 s per sample. The sensor was applied to detect cell viability with inserting PHMG-p as toxic matter and we found NADH level is increased when insert high level of PHMG-p because it caused cell death. This demonstrates NADH electrocatalytic sensor will become a powerful tool for cell viability assay and pave to the way to toxicological study.
Reviewer 2 Comment 7: Graphical abstract. It can be provided.
Response and Action: We thank the reviewer for his/her criticism. We made the graphical abstract and insert the manuscript. Thank you.
Graphical Abstract (insert it at last page)
Figure. The electrocatalytic reaction in cell culture medium to detect NADH.
Reviewer 2 Comment 8: Papers cited here should be updated. Only 2 papers published in recent 3 years were cite.
Response and Action: We thank the reviewer for his/her criticism. We updated the paper in recent 5 years.
Insert table 1,
Table 1. Comparison of NADH detection with various analytical methods.
Yours Sincerely,
Dr. JuKyung Lee
Research Professor
Medical IT Convergence Engineering
Kumoh National Institute of Technology
Gumi, 39177, Korea
Tel: +82-54-478-6995
Fax: +82-54-478-6990
E-mail: chejueyes@kumoh.ac.kr

Round 2
Reviewer 2 Report
Accept in present form